# Radiomics profiling combined with clinical risk factors for preoperative Lymphatic Metastasis prediction in Colorectal cancer: A multicenter study

Fangda Guo[1☯], Jingru Li[1☯]*, Liang Chen[2], Jie Hu[1], Liezhen Wang[1], Wenyan Gu[1], Lijing Liu[3]

**1** Medical Equipment Center, Ningbo Medical Center Lihuili Hospital, Yinzhou District, Ningbo, Zhejiang, China, **2** Radiographic Imaging Department, Ningbo Medical Center Lihuili Hospital, Yinzhou District, Ningbo, Zhejiang, China, **3** Department of Radiation Oncology, Changxing Branch, The Second Affiliated Hospital of Zhejiang University School of Medicine, Huzhou, Zhejiang, China

☯ Fangda Guo and Jingru Li contributed equally to this work.
* LHLlijingru@outlook.com

## Abstract

### Purpose

Accurate preoperative assessment of regional lymphatic metastases (LNM) is essential for effective surgical selection of patients with colorectal cancer (CRC). This study aimed to develop a machine learning (ML) model that integrates radiomics and clinical risk factors to predict preoperative LNM in CRC patients.

### Methods

This multicenter cohort study retrospectively collected data from 349 CRC patients between January 1, 2020, and December 31, 2023. A total of 292 patients from our hospital comprised the training dataset, while 57 patients from external hospitals formed the validation dataset. Radiomic features of the tumor region (3D(R)) and colorectal region (3D(C)) were extracted from venous-phase CT images. LASSO (least absolute shrinkage and selection operator) regression was applied to screen clinical and radiomic features. 4 prediction models, clinical, 3D(R), 3D(R+C), and combined, were constructed using support vector machine (SVM). The optimal model was identified through comparative analysis of the area under the curve (AUC) metric across multiple models.

### Results

The Model_3D(R+C) demonstrated superior discriminative performance compared to Model_3D(R) alone (AUC: training, 0.733(95% CI: [0.693, 0.773]) vs. 0.696 (95% CI: [0.655, 0.737]); validation, 0.641(95% CI: [0.590, 0.692]) vs. 0.563(95%

**Data availability statement:** All relevant data are in the manuscript and its supporting information files.

**Funding:** Ningbo Medical Center Lihuili Hospital: Ningbo, CN. The funders provided full support in the data collection and analysis efforts.

**Competing interests:** The authors have declared that no competing interests exist.

CI: [0.507, 0.619])). The model combining clinical and 3D(R + C) (ModelC_3D(R + C)) outperformed the clinical model(ModelC) and Model_3D(R + C) (AUC: training: 0.858(95% CI: [0.826, 0.890]) vs. 0.635(95% CI: [0.585, 0.685]) vs. 0.733(95% CI: [0.693, 0.773]); validation 0.833(95% CI: [0.787, 0.879]) vs. 0.589(95% CI: [0.537, 0.641]) vs. 0.641(95% CI: [0.590, 0.692]); $P < 0.050$). Therefore, the combined model provided the most accurate identification of LNM.

## Conclusion

The SVM model incorporating 3D(R) features, 3D(C) features, and clinical risk factors effectively predicts preoperative LNM in CRC patients.

## Introduction

CRC is a highly aggressive malignancy impacting the digestive system. As of 2022, CRC ranked as the third most prevalent cancer worldwide, accounting for 9.6% of all new cancer diagnoses. Additionally, CRC was responsible for 9.3% of all cancer-related deaths, making it the second leading cause of cancer mortality globally [1,2]. The treatment approach for CRC patients is guided by TNM staging, which assesses the primary tumor (T), lymph node involvement (N), and distant metastasis (M) [3]. Numerous studies indicate that preoperative LNM status is a crucial factor in determining the surgical plan and scope, significantly influencing postoperative survival and prognosis in CRC patients [4–6].

Research indicates that patients with high-risk factors constitute approximately 70–80% of the early-stage CRC patient population. However, LNM is detected in only about 10% of early-stage patients, suggesting that approximately 60% of these patients may be overtreated [7]. Thus, current methods for preoperative LNM assessment lack accuracy. The tendency to overestimate LNM risk has led to unnecessary surgeries, subjecting many patients to surgical trauma and increasing healthcare costs. To enhance precision at the individual level and reduce overtreatment, there is a critical need for predictive models capable of accurately assessing early colorectal LNM in clinical settings.

Currently, preoperative diagnosis of CRC primarily relies on conventional imaging techniques, including computed tomography (CT) and magnetic resonance imaging (MRI) [8]. These modalities offer anatomical and morphological insights into the colorectal region, supporting diagnostic processes. However, these methods have limited predictive accuracy forLNM. According to the American Society of Colon and Rectal Surgeons (ASCRS), CT has a sensitivity of 55% and a specificity of 74% in detecting regional LNM in rectal cancer, while MRI shows a sensitivity of 66% and specificity of 76%. These findings suggest that conventional imaging techniques are inadequate for precise regional LNM prediction inCRC. Conventional imaging-based diagnosis primarily depends on radiologists' direct observations, which are influenced by their expertise, experience, and the quality of imaging equipment. Furthermore, patient-specific factors, including the unique anatomical characteristics

of each patient's colorectal region, may impact radiologists' interpretations. Currently, the risk of both over-treatment and under-treatment in CRC patients remains a concern, potentially impacting patient outcomes and prognosis. Consequently, there is a critical need for an approach capable of accurately predicting preoperative LNM in CRC patients.

Radiomics, an interdisciplinary field combining radiology and molecular biology, focuses on extracting quantitative features from medical images using automated data extraction algorithms. The extracted features are subsequently analyzed by computational methods, particularly ML, to model and interpret the underlying histological characteristics. This approach facilitates clinical decision-making and supports the advancement of precision medicine [9–11]. The combined application of radiomics and ML has demonstrated potential in accurately identifying LNM [12,13].

However, ML models based on individual radiomic features often face limitations due to the constrained information in available image data. Studies have demonstrated that several clinical risk factors substantially contribute to LNM. Computed tomography (CT) is the most widely used and cost-effective diagnostic tool for CRC patients [14,15]. In conventional imaging analysis, the tumor region of interest (ROI) is typically segmented to extract histological features. However, recent studies suggest that imaging characteristics of surrounding tissues may also influence LNM [16–18]. Therefore, this study aims to predict preoperative LNM by integrating radiomic features of the tumor and surrounding tissues from CT images with clinical risk factors. This integrated approach has the potential to significantly improve diagnostic accuracy. Model predictive performance is typically evaluated using sensitivity, specificity, and area under the curve (AUC). Models with an AUC above 0.9 are considered to exhibit very high diagnostic efficacy, while those with AUC values between 0.7 and 0.9 show good diagnostic efficacy. AUC values below 0.7 suggest limited diagnostic efficacy. This study aims to develop a model with an AUC exceeding 0.7, which would support its clinical significance.

## Materials and methods

### Patient selection

This retrospective study collected data from patients diagnosed with CRC via postoperative pathology reports at LHL Hospital and HZP Hospital, covering the period from January 1, 2020, to December 31, 2023, with data collection initiated on July 10, 2024. The dataset included whole-abdominal enhanced CT images, clinical laboratory data, and pathology reports. Approximately 10% of these cases involved rectal cancer. Since adenocarcinoma and mucinous adenocarcinoma are the predominant pathohistological types of CRC, our patient selection focused primarily on these two types. This selection criterion ensured the relevance and applicability of our findings, particularly for preoperative LNM prediction.

Inclusion criteria: 1.Age ≥ 18 years; 2.Confirmed CRC diagnosis via postoperative pathology report; 3.Enhanced CT scan of the upper or whole abdomen within two weeks before surgery; 4.Availability of complete case information, including age, gender, presence of comorbidities, and any preoperative treatments; 5.Complete physiological and biochemical laboratory results within one week before surgery, including three tumor markers: carbohydrate antigen (CA199), alpha-fetoprotein (AFP), and carcinoembryonic antigen (CEA).

Medical history exclusion criteria: 1.History of prior surgeries for other cancers; 2.Prior adjuvant radiotherapy before surgery; 3.Poor image quality or unlabeled lesion location;4.History of colorectal stent placement.

According to the relevant literature, regional lymph nodes commonly affected in CRC include the superior and inferior mesenteric lymph nodes, lymph nodes along the abdominal aorta and iliac arteries, as well as perineal and rectal lymph nodes [19]. These lymph nodes are crucial for CRC treatment and pathological staging. In this study, the postoperative tumor pathology report served as the reference standard for model labeling. LNM was labeled as LNM+ if present and as LNM- if absent. no LNM was indicated as LNM-.

A total of 349 patients were included in the study. The study protocol received approval from the hospital's ethical review board (approval number: KY2024SL221−01). Since patient data were collected retrospectively, informed consent was waived. All patient data were anonymized to protect individual privacy. The methodology for patient selection is illustrated in Fig 1.

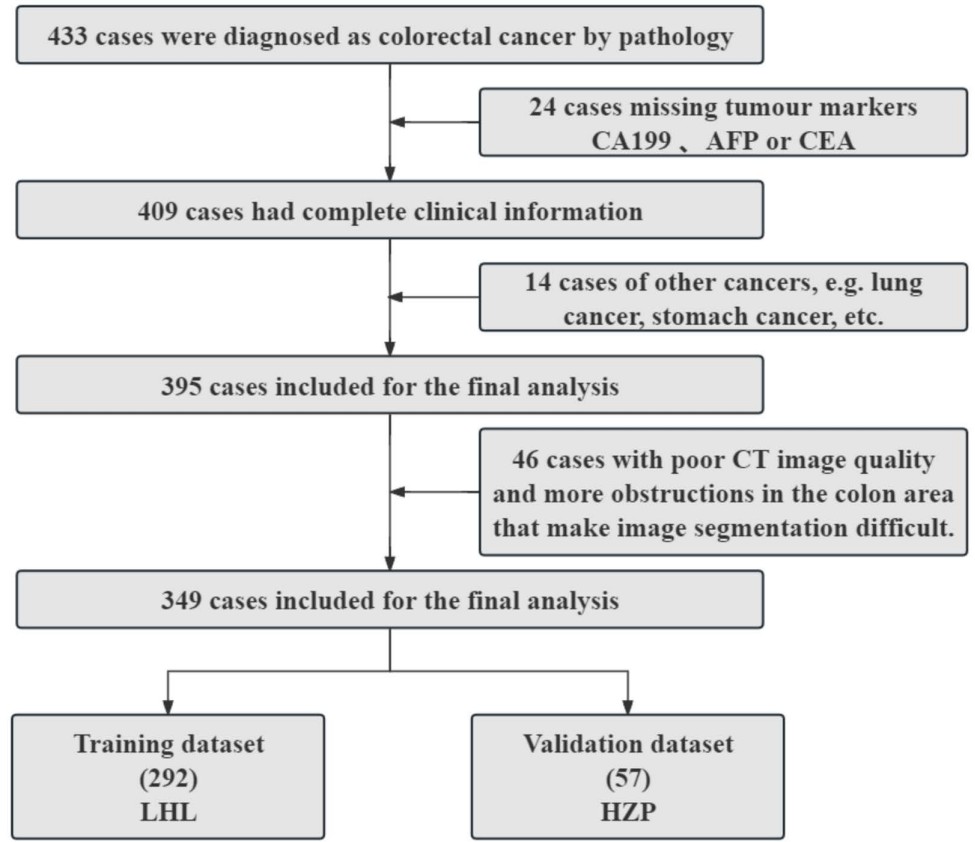

**Fig 1. Inclusion and exclusion flowcharts.**

## Image acquisition and 3D segmentation

To ensure consistency between the training and validation datasets, several pre-processing steps were applied to the CT images. 1.CT Image Selection: Only venous-phase CT images were selected for both the training and validation datasets, ensuring consistency in the imaging phase across all cases. 2.Image Resizing and Dimensions: All CT images were resized to a consistent dimension of 512x512 pixels to ensure uniformity in image resolution. 3.Image Slice Thickness: A slice thickness of 5 mm was applied to all images, both in the training and validation datasets, to maintain uniformity in the 3D reconstruction of the tumors. 4.Image Windowing: During the image segmentation process, the Level and Window values were set to 100 and 300, respectively, for all images, standardizing the contrast and brightness across the datasets.These pre-processing steps were uniformly applied to both the training and validation datasets to ensure that the data used for model training and validation were harmonized and comparable.In this study, the colorectal region of interest (ROI) was manually segmented to precisely delineate the tumor area, excluding surrounding structures such as lymph nodes. The segmentation was performed using ITK-SNAP software, with the primary objective of isolating the tumor within the colorectal region. The tumor area was delineated to capture its full extent while avoiding the inclusion of normal bowel tissue or other non-tumor structures. Venous-phase CT images, which provide superior clarity in distinguishing tumor tissue from surrounding structures, guided the segmentation process. Given the well-defined appearance of colorectal tumors on CT images, this manual segmentation approach was both reproducible and effective in accurately capturing the tumor region. To ensure consistency and accuracy, two senior gastrointestinal radiologists reviewed and confirmed all

manually segmented tumor regions, verifying segmentation accuracy and ensuring that no additional structures, such as lymph nodes or paracolic tissue, were included in the tumor ROI. To evaluate the reproducibility of ROI segmentation, we conducted intra-class correlation coefficient (ICC) analyses. The ICC for segmentation performed by the same observer on the same patient one day apart was 0.973, indicating excellent intra-observer reliability. Furthermore, the ICC calculated between two different observers segmenting the same patient's ROI was 0.972, demonstrating strong inter-observer agreement. These results further support the reliability of our radiomic feature extraction process. The colorectal region in this study refers to the anatomical area from the cecum to the rectum, encompassing the colon, rectum, and surrounding paracolic tissue. Key anatomical landmarks, such as the segments of the colon and the upper and lower ends of the rectum, define the boundaries of this region. Given the region's extensive size, manual segmentation may introduce notable errors. Therefore, an automated tool was employed to segment the colorectal region, ensuring comprehensive imaging of this region. The segmentation method was derived from the Total Segmentator tool (https://totalsegmentator.com/r). Images obtained with this tool were evaluated and confirmed by two senior imaging physicians, with similarly satisfactory reproducibility. The agreement (intra-group correlation coefficient, ICC) between physician-annotated images and tool-generated images was 0.994, indicating strong concordance. Multiple automated segmentations of the same patient's image using the tool yielded an ICC of 0.989, indicating strong reproducibility.

## Data extraction

Data extraction was divided into two main parts: (1) clinical data, including three primary tumor markers (CEA, CA199, and AFP), blood counts, and biochemical test results. A review of the literature indicates that tumor marker parameters play a significant role in predicting postoperative prognosis for cancer patients. [20,21](Clinical factors were statistically analyzed in S1 Table). A binary classification was applied to the tumor markers CA199 and CEA, where a value of 0 indicated a negative result, and a value of 1 indicated a positive result. Routine blood and biochemical tests evaluate a patient's overall health and surgical suitability, offering valuable data to potentially enhance the predictive model. Thus, these clinical parameters were incorporated into our data collection. Radiomic analysis captures high-dimensional features within CT images that are beyond visual discernment, providing a rich data source for clinical decision-making. [21–23]. Radiomic features were extracted from both tumor (R) and colorectal (C) regions using the PyRadiomics package in Python (version 3.1.0). A total of 1,240 3D (R) features and 1,240 3D (R+C) features were extracted, encompassing histograms, textures, and additional relevant characteristics. All analyses and model implementations were conducted within a Python 3.7 environment. In this study, the presence of LNM indicated in pathology reports served as the primary label for preoperative prediction. Consequently, other details in the pathology report, such as tumor deposits, were deemed to have minimal relevance to our analysis and results, and thus were excluded from data collection.

## Statistical analysis

To evaluate the model's generalization capacity, external data were used for validation in this study. A total of 292 patients from LHL Hospital were assigned to the training dataset, while 57 patients from HZP Hospital made up the validation dataset. Quantitative data were expressed as either the mean (standard deviation) or the median (interquartile range), depending on data distribution. Distributions between groups were compared using the t-test or the Wilcoxon rank-sum test. Categorical variables were presented as percentages and compared using Pearson's chi-square test or Fisher's exact test.

First, we divided the patient data into a training set and an external validation set. After extracting features using radiomic methods, we performed 0–1 normalization on all extracted features due to significant differences in the ranges of radiomic feature values caused by various scanning devices and parameter settings. These differences could potentially affect the evaluation of tumor characteristics. This normalization process is a crucial step to ensure the comparability of features and the reliability of the analysis results. Then, LASSO was applied for the initial screening of clinical and 3D

(R + C) features, followed by further screening using grid search and tenfold cross-validation methods. Third, SVM was used to select (C) features, (R) features, and a combination of both, to construct Model_3D(R) and Model_3D(R + C), respectively. This study compares the AUC values of Model_3D(R) and Model_3D(R + C) to illustrate the advantages of integrating tumor and colorectal features in a combined modeling approach. Fourth, it was determined that 3D(R + C) features exhibit superior modeling capabilities. Consequently, clinical features from the screening were combined with 3D(R + C) features to construct two models: a single clinical model (ModelC) and a combined clinical model (ModelC_3D(R + C)). The performances of the three models—ModelC, Model_3D(R + C), and ModelC_3D(R + C)—were then compared. Ultimately, AUC and decision curve analysis (DCA) were conducted for the three models. AUC serves as a critical metric for evaluating dichotomous model performance, while DCA is an innovative method for assessing the clinical utility of predictive models, integrating sensitivity, specificity, and expected net benefit across different risk thresholds to enable a comprehensive model evaluation. Consequently, the optimal model was identified.

All feature extraction and model-building algorithms were implemented in Python (version 3.7). Statistical analyses and feature extraction were conducted using R (version 4.2.1). SVM was implemented with the "scikit-learn" package (https://scikit-learn.org/stable/). All statistical tests were two-sided, with p-values ≤ 0.05 considered statistically significant.

## Results

### Study population and baseline

A total of 349 patients were included in this study, categorized into a training dataset (292 from LHL Hospital) and an external validation dataset (57 from HZP Hospital). No statistically significant differences were observed between the training and validation datasets in terms of demographic factors. Table 1 presents the baseline characteristics of the patients. A total of 139 patients had postoperative pathology reports showing LNM+ status (110 in training; 29 in validation) (Table 1).

**Table 1. Baseline demographics of patients.**

| Clinical factors | Training dataset (N = 292) | Validation dataset (N = 57) | p Value |
|---|---|---|---|
| **Age (year)** | 64.4 ± 11.1 | 65.1 ± 11.9 | 0.699 |
| **Sex (N)** | | | 0.770 |
| Male | 172 | 35 | |
| Female | 120 | 22 | |
| **Swollen lymph nodes** | | | 0.001* |
| YES | 146 | 11 | |
| NO | 146 | 46 | |
| **AFP ug/L** | 2.87(2.0,3.4) | 2.77(1.8,3.4) | 0.622 |
| **CEA ug/L** | | | 0.561 |
| <5.0 | 179 | 32 | |
| ≥5.0 | 113 | 25 | |
| **CA199 U/ml** | | | 0.834 |
| <37.0 | 250 | 50 | |
| ≥37.0 | 42 | 7 | |
| **Postoperative lymph nodes Transfer case** | | | 0.086* |
| YES | 110 | 29 | |
| NO | 182 | 28 | |

Normally distributed factors are expressed using means ± standard deviations; non-normally distributed factors are expressed as medians (interquartile ranges)

*With a p < 0.050

## Model construction

In this study, an initial total of 20 clinical features, along with 1240 colon and 1240 region of interest (ROI) features from imaging histology, were considered. After LASSO screening, the final feature set included 3 clinical features, 2 3D (C) features, and 4 3D (R) features, consisting of 1 morphological factor and 4 high-dimensional factors (S2 Table).

## Comparison between the Model3D(R + C) and Model_3D(R)

A comparison between Model_3D(R + C) and Model_3D(R) showed that the AUC of Model_3D(R + C) was higher than that of Model_3D(R) in both the training dataset (AUC = 0.733 (95% CI: [0.693, 0.773]) vs. 0.696(95% CI: [0.655, 0.737]), Fig 2A) and the validation dataset (AUC = 0.641 (95% CI: [0.590, 0.692]) vs. 0.563(95% CI: [0.507, 0.619]), Fig 2B). Based on these findings, we intend to employ the 3D(R + C) features in subsequent modeling processes.

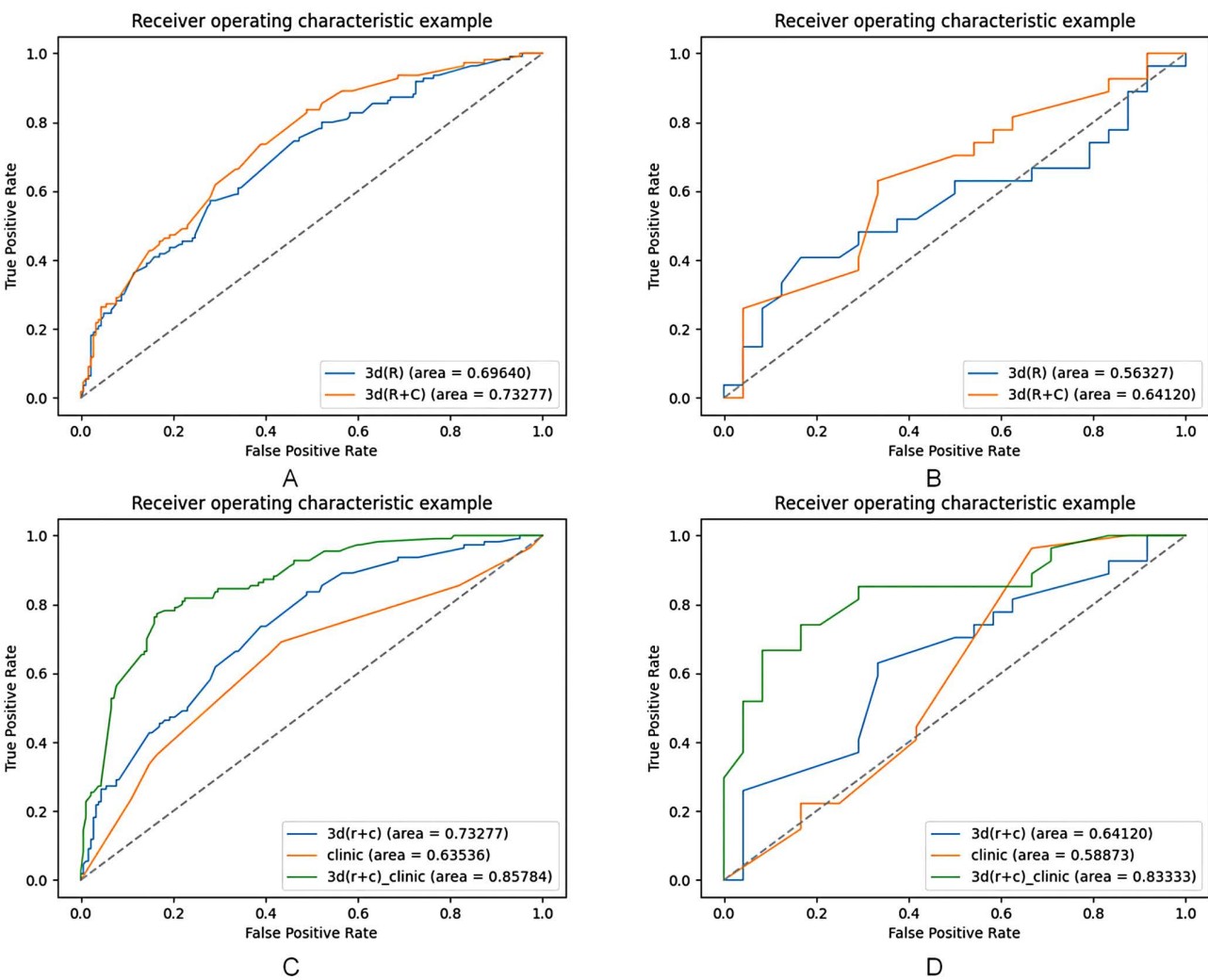

**Fig 2. AUC analysis for each model.** A and B show the training set vs. validation set AUC for model 3D(R) vs. model 3D(R + C). C and D show the training set vs. validation set AUC for model 3D(R + C), the clinical model vs. the combined model.

## Comparison between the Model3D(R + C),ModelC and ModelC_3D(R + C)

A comparison of Model_3D(R + C), ModelC, and ModelC_3D(R + C) reveals that ModelC_3D(R + C) demonstrates superior performance in terms of AUC on both datasets: training (AUC = 0.858(95% CI: [0.826, 0.890]) vs. 0.733 (95% CI: [0.693, 0.773]) vs. 0.635 (95% CI: [0.585, 0.685]), Fig 2C) and validation (AUC = 0.833(95% CI: [0.787, 0.879]) vs. 0.641 (95% CI: [0.590, 0.692]) vs. 0.589(95% CI: [0.537, 0.641]), Fig 2D). The DCA curves are presented in Fig 3A. Based on these results, ModelC_3D(R + C) was identified as the optimal model. For this model, feature contributions were analyzed using

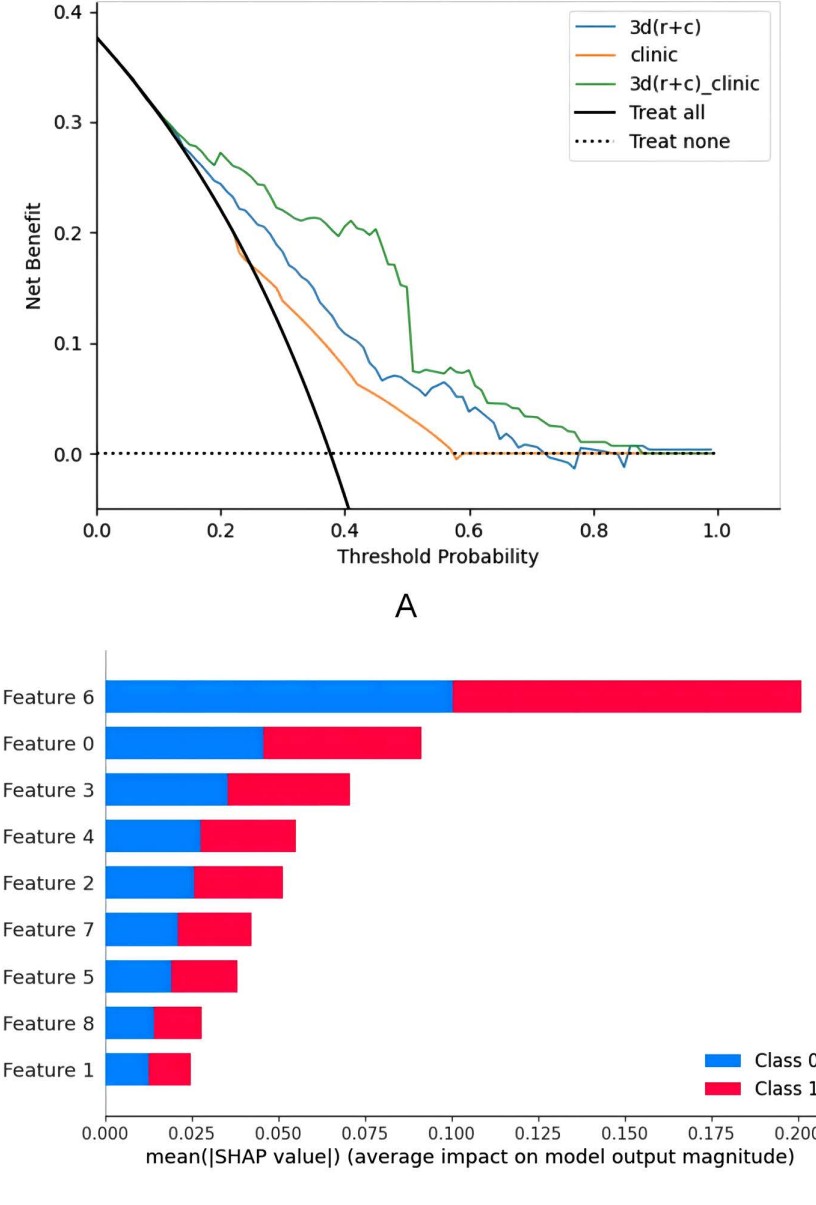

**Fig 3. DCA and SHAP effect diagrams for combined models.** A Decision Curve Analysis(DCA) compares the combined model with two other models. B demonstrates the contribution of each feature in the combined model, with the specific name of each feature in the Supplementary Table.

SHapley Additive exPlanations (SHAP), as shown in Fig 3B. The AUC, sensitivity, and specificity of the three models were then compared (S4 Table).

### Comparison between the combined model and the basic models (Mean, standard deviation, and size of the tumor ROI)

A comparison of the combined model with the models based on mean, standard deviation, and tumor ROI size shows that the combined model demonstrates superior performance in terms of AUC on both datasets:Training set (AUC = 0.858 vs. 0.575 vs. 0.541 vs. 0.579); Validation set (AUC = 0.833 vs. 0.426 vs. 0.467 vs. 0.444). These results highlight that the combined model significantly outperforms the models built directly from basic parameters (Fig 4).

## Discussion

The status of regional LNM in CRC is crucial for determining adjuvant therapy and surgical resection strategies. This study aimed to evaluate the predictive value of 3D-enhanced CT radiographic features and clinical risk factors for preoperative LNM in CRC patients. Most prior studies have focused on extracting histologic features from the lesion area for LNM prediction. In contrast, this study adopted a more comprehensive approach, incorporating not only three-dimensional radiographic features from the tumor region but also broader colorectal features wherever feasible. These features were then modeled alongside clinical risk factors to develop a robust predictive model for LNM in CRC patients [24]. Predicted LNM in CRC patients using both plain and venous-phase enhanced CT images, achieving AUCs of 0.636 and 0.690, respectively. Therefore, this study derived radiological features exclusively from venous-phase enhanced CT images, excluding plain CT images from feature extraction.

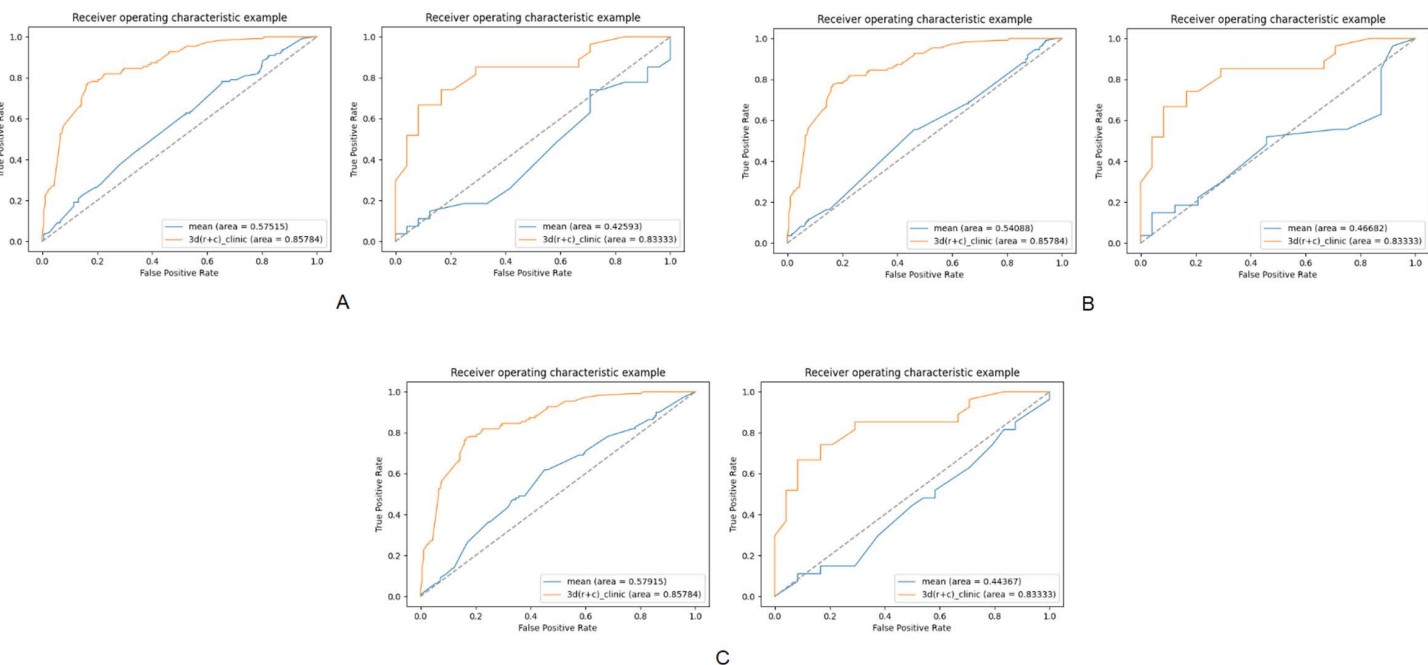

**Fig 4. AUC analysis for each model.** Panels A, B and C present the AUC comparisons of the combined model, mean model, standard deviation model, and tumor ROI size model for both the training and validation sets. In each panel, the left image shows the AUC for the training set, while the right image shows the AUC for the validation set.

Radiomics is a technique that extracts numerous high-throughput features from medical images, including those from CT, MRI, PET, and other modalities. These features include various characteristics, such as shape and texture. By processing and analyzing these features, information that is not discernible to the naked eye can be obtained. Furthermore, analyzing the entire tumor volume helps eliminate bias errors similar to those found in pathology sampling, offering a potential advantage of radiomics analysis [25]. In conventional radiomics, feature values are typically extracted from the tumor region for analytical modeling. However, studies have shown that the surrounding region may also contain features that influence prediction outcomes. Therefore, this study extracted features not only from the tumor region but also from the surrounding colorectal region.

Our model was developed using venous-phase CT alone to prioritize availability and clinical practicality. Although we initially considered incorporating MRI/PET features, in routine CRC practice contrast-enhanced CT is most widely accessible, whereas MRI and PET are not uniformly available across hospitals. Nonetheless, MRI (e.g., T2-weighted, DWI/ADC) and PET (e.g., SUV-based metrics) may provide complementary functional and metabolic information that could enhance performance. We are currently engaging in research exchanges with multiple centers with the aim of collecting datasets from CRC patients who have CT, MRI, and PET imaging available, upon which we will explore developing a higher-performing multimodal predictive model.

We extracted a total of 1,240 tumor image features and 1,240 colorectal region features. After LASSO screening, four tumor image features were retained. Three of these were high-order grayscale features derived from transformations such as wavelet and Gaussian filtering: log.sigma.3.mm.3D_glszm_GrayLevelNonUniformity, wavelet.HLH_glcm_InverseVariance, and wavelet.LHH_glszm_SmallAreaHighGrayLevelEmphasis. The fourth feature reflects the compactness of the tumor area in the original image, capturing texture variations within the tumor. This indicates a possible strong correlation between tumor heterogeneity and the metastatic profile of LNM, as radiomic features like GLSZM, GLCM, and GLDM are often associated with structural heterogeneity and irregularity [26]. In addition to the four tumor image features, we retained two colorectal region features: log.sigma.3.mm.3D_gldm_SmallDependenceHighGrayLevelEmphasis and log.sigma.3.mm.3D_glszm_LowGrayLevelZoneEmphasis. Both features represent texture characteristics specific to the colorectal region, suggesting that colorectal texture may also influence LNM status.

Based on the SHAP values obtained from our analysis, we found that the original_shape_Compactness and log-sigma_3_mm_3D_gldm_SmallDependenceHighGrayLevelEmphasis features significantly contribute to our predictive model for lymph node metastasis. The former reflects tumor geometric properties, suggesting that higher compactness may correlate with lower metastatic potential. In contrast, the latter emphasizes high gray-level areas that are indicative of increased vascularity, thus highlighting the importance of tumor biology in predicting lymph node involvement. By elucidating these associations, we aim to provide clinicians with a clearer understanding of how specific imaging features relate to tumor biology.

As expected, Model_3D(R + C) demonstrated superior predictive accuracy compared to Model_3D(R) on the validation set (AUC: 0.641 (95% CI: [0.590, 0.692]) vs. 0.563 (95% CI: [0.507, 0.619])). However, Model_3D(R + C) lacked the required resolution for clinical predictability (training: 0.733 (95% CI: [0.693, 0.773]); validation: 0.641(95% CI: [0.590, 0.692])). In clinical practice, models with an AUC greater than 0.7 are generally considered to have clinical relevance [7].

In addition to radiological features, certain clinical risk factors may also influence the occurrence of LNM. During this study, three clinical features were identified through LASSO screening, with CEA being a commonly used tumor marker. CEA levels are markedly elevated in CRC patients, making this biomarker a crucial tool for cancer detection and prognosis. In contrast, CEA levels are typically low in normal cells, making it a more specific cancer marker [27,28]. However, the specificity of CEA is insufficient for individual modeling, as shown by the model's poor performance when CEA was used alone with other clinical risk factors. As shown by the results of ModelC (AUC: training: 0.635 (95% CI: [0.585, 0.685]); validation: 0.589 (95% CI: [0.537, 0.641])), these values are much lower than 0.7 and are not clinically predictive. Therefore, a single clinical risk factor cannot serve as the sole eigenvalue for modeling. However, according to relevant

studies, clinical risk factors such as CEA still hold clinical diagnostic value and can serve as an adjunct to clinical diagnosis [29]. In this study, clinical risk factors were combined with 3D(R + C) features in the modeling process. The combined ModelC_3D(R + C) demonstrated a significant improvement in discriminative ability (AUC: training: 0.858 (95% CI: [0.826, 0.890]); validation: 0.833 (95% CI: [0.787, 0.879])). This result suggests that combining clinical factors with 3D(R + C) features is an effective approach for predicting preoperative LNM. The AUC of ModelC_3D(R + C) exceeds 0.8, indicating that our combined model demonstrates excellent predictive performance and, consequently, has substantial clinical significance in predicting preoperative LNM.

ML is a branch of artificial intelligence focused on enabling computer systems to learn patterns and regularities from data and subsequently use this knowledge to make decisions. The core principle involves training models with data, enabling them to make accurate predictions and judgments about new data. Selecting the most suitable modeling algorithms for specific data types is crucial, as different algorithms yield models with varying predictive performance. In this study, we evaluated five models: SVM, Random Forest, XGBoost, Logistic Regression, and Decision Tree. Clinical factors and three-dimensional (R + C) features were modeled separately using five different algorithms (Training: 0.858(95% CI: [0.790, 0.876]) vs. 0.839 (95% CI: [0.800, 0.878]) vs. 0.803(95% CI: [0.762, 0.844]) vs. 0.772(95% CI: [0.730, 0.814]) vs. 0.783 (95% CI: [0.741, 0.825]) (S1A Fig); Validation: 0.833 (95% CI: [0.790, 0.876]) vs. 0.562 (95% CI: [0.511, 0.613]) vs. 0.533(95% CI: [0.482, 0.584]) vs. 0.613 (95% CI: [0.570, 0.656]) vs. 0.572 (95% CI: [0.521, 0.623]) (S1B Fig)). The best modeling results were achieved using SVM. SVM maps the data into a high-dimensional space using a kernel function, making it well-suited for datasets with high feature dimensions and enabling nonlinear classification. SVM generally demonstrates superior generalization ability [30]. These findings suggest that integrating SVM with radiomics and clinical variables enhances the accuracy of preoperative LNM prediction in CRC patients. This approach has the potential to improve treatment planning precision, ultimately contributing to better patient survival and prognosis. Therefore, the model developed in this study carries significant clinical implications.

The false positive rate (FPR) is an essential metric to evaluate the reliability of the model in clinical practice. In our study, the combined model achieved a relatively low FPR of 8.33% in the validation cohort, indicating that the model performed well in distinguishing tumor from non-tumor cases and yielding high specificity (~91.7%) alongside good overall discrimination (AUC > 0.83). However, the sensitivity at the chosen operating point remained moderate (55.6%), implying a risk of false negatives (missed LNM) that warrants cautious interpretation, particularly when LNM status could alter management. This operating profile is conservative—helping to reduce unnecessary interventions and avoid over-diagnosis—yet it underscores the need to tailor decision thresholds to clinical priorities: lowering the threshold to increase sensitivity when avoiding under-treatment is critical, and raising the threshold to emphasize specificity when minimizing over-treatment is paramount. For routine use, a balanced operating point (e.g., near the Youden index) may provide an appropriate trade-off. We recommend site-specific calibration and the use of calibration assessment and decision-curve analysis to identify thresholds that maximize net benefit. Probability-based risk stratification (e.g., defining low-, intermediate-, and high-risk bands) may further support nuanced decision-making. Although some misclassifications were observed, the model demonstrated a favorable balance among sensitivity, specificity, and FPR, and is intended to complement—rather than replace—clinical judgment in predicting preoperative LNM in CRC patients.

This study is subject to several limitations. First, the sample size is relatively small. Second, It should be noted that the external validation cohort was small (n = 57), which may lead to wider confidence intervals around performance estimates and therefore warrants cautious interpretation. Although the two-center design provides preliminary evidence of generalizability, broader multicenter validation is necessary to better capture heterogeneity across institutions. At present, we are engaging in research exchanges with multiple centers and hope to collect and utilize multicenter data for external validation in subsequent work. Although standardized imaging protocols were used, differences in scanner models, image resolution, and acquisition techniques could still introduce variability in the quality of images, which may affect the consistency of feature extraction and model performance across centers. Third, the radiomics features in this study were

extracted from the region of interest (ROI) using ML. However, the segmentation of the ROI is manually designed, relying on the expertise of domain specialists, which introduces challenges in feature extraction and limits flexibility. Segmentation variability can propagate to radiomics features via boundary placement, ROI size/shape, and the inclusion or exclusion of adjacent tissues. Boundary shifts predominantly affect shape metrics (e.g., volume, surface area, sphericity/compactness), boundary-sensitive texture features (e.g., GLCM contrast/entropy, GLRLM/GLSZM nonuniformity), and high-frequency wavelet or Laplacian-of-Gaussian features, whereas global intensity histogram features may be less sensitive to small boundary changes. Partial-volume effects at the tumor–bowel interface and inadvertent inclusion of peritumoral tissues can alter heterogeneity descriptors and impact downstream model outputs. To mitigate variability, tumor ROIs were manually delineated on venous-phase CT and reviewed by two radiologists; automated colorectal segmentation based on Total Segmentator showed excellent agreement with expert annotations (ICC = 0.994) and good repeatability (ICC = 0.989), supporting the stability of features in this region. In future work, we will explore deep learning–driven tumor auto-segmentation to further reduce operator dependence and enhance cross-center stability.

In conclusion, the developed model demonstrates enhanced accuracy in predicting preoperative LNM in CRC patients. By leveraging this model, clinicians can devise more precise treatment plans, reducing the risks of both under-treatment and over-treatment. This approach minimizes unnecessary harm to patients and significantly improves post-surgical therapeutic efficacy and quality of life.

## Supporting information

**S1 Table. Baseline statistics of LNM+ versus LNM- patient population.**
(DOC)

**S2 Table. Modeling factors.**
(DOC)

**S3 Table. Pairwise comparison of the models.**
(DOC)

**S4 Table. Comparison of AUC, sensitivity, and specificity of the models.**
(DOC)

**S1 Fig. AUC plots of validation set and training set for the five models.**
(TIF)

**S1 File. Image feature data for the validation.**
(CSV)

**S2 File. Clinical characterization data for the validation.**
(CSV)

**S3 File. Image feature data for the training.**
(CSV)

**S4 File. Clinical characterization data for the training.**
(CSV)

## Acknowledgments

We thank all authors for their help in this study.

## Author contributions

**Conceptualization:** Fangda Guo.

**Data curation:** Jingru Li, Liang Chen.

**Formal analysis:** Liang Chen, Liezhen Wang.

**Funding acquisition:** Fangda Guo, Liang Chen.

**Investigation:** Wenyan Gu, Lijing Liu.

**Methodology:** Fangda Guo, Jingru Li, Liang Chen.

**Project administration:** Jingru Li, Liang Chen.

**Resources:** Fangda Guo, Jingru Li, Jie Hu, Liezhen Wang, Lijing Liu.

**Software:** Fangda Guo, Jingru Li.

**Supervision:** Fangda Guo.

**Validation:** Fangda Guo, Jie Hu.

**Visualization:** Wenyan Gu.

**Writing – original draft:** Jingru Li, Jie Hu.

**Writing – review & editing:** Fangda Guo, Jingru Li.

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
