## [Decision Letter · Decision Letter 0]

22 Oct 2024

Dear Dr. Li,

Thank you for submitting your manuscript to PLOS ONE. After careful consideration, we feel that it has merit but does not fully meet PLOS ONE’s publication criteria as it currently stands. Therefore, we invite you to submit a revised version of the manuscript that addresses the points raised during the review process.

We look forward to receiving your revised manuscript.

Kind regards,

Paolo Aurello

Academic Editor

PLOS ONE

Journal requirements: When submitting your revision, we need you to address these additional requirements. 1. Please ensure that your manuscript meets PLOS ONE's style requirements, including those for file naming. The PLOS ONE style templates can be found at https://journals.plos.org/plosone/s/file?id=wjVg/PLOSOne_formatting_sample_main_body.pdf and https://journals.plos.org/plosone/s/file?id=ba62/PLOSOne_formatting_sample_title_authors_affiliations.pdf 2. Please note that PLOS ONE has specific guidelines on code sharing for submissions in which author-generated code underpins the findings in the manuscript. In these cases, all author-generated code must be made available without restrictions upon publication of the work. Please review our guidelines at https://journals.plos.org/plosone/s/materials-and-software-sharing#loc-sharing-code and ensure that your code is shared in a way that follows best practice and facilitates reproducibility and reuse. 3. We note that the grant information you provided in the ‘Funding Information’ and ‘Financial Disclosure’ sections do not match.  When you resubmit, please ensure that you provide the correct grant numbers for the awards you received for your study in the ‘Funding Information’ section. 4. Thank you for stating the following financial disclosure:  [Ningbo Medical Center Lihuili Hospital: Ningbo, CN].  Please state what role the funders took in the study.  If the funders had no role, please state: ""The funders had no role in study design, data collection and analysis, decision to publish, or preparation of the manuscript."" If this statement is not correct you must amend it as needed. Please include this amended Role of Funder statement in your cover letter; we will change the online submission form on your behalf. 5. Thank you for stating the following in the Acknowledgments Section of your manuscript: [We thank Li Huili Hospital of Ningbo Medical Center and all authors for their help in this study.]We note that you have provided funding information that is not currently declared in your Funding Statement. However, funding information should not appear in the Acknowledgments section or other areas of your manuscript. We will only publish funding information present in the Funding Statement section of the online submission form. Please remove any funding-related text from the manuscript and let us know how you would like to update your Funding Statement. Currently, your Funding Statement reads as follows:  [Ningbo Medical Center Lihuili Hospital: Ningbo, CN] Please include your amended statements within your cover letter; we will change the online submission form on your behalf. 6. We note that you have indicated that there are restrictions to data sharing for this study. For studies involving human research participant data or other sensitive data, we encourage authors to share de-identified or anonymized data. However, when data cannot be publicly shared for ethical reasons, we allow authors to make their data sets available upon request. For information on unacceptable data access restrictions, please see http://journals.plos.org/plosone/s/data-availability#loc-unacceptable-data-access-restrictions.  Before we proceed with your manuscript, please address the following prompts: a) If there are ethical or legal restrictions on sharing a de-identified data set, please explain them in detail (e.g., data contain potentially identifying or sensitive patient information, data are owned by a third-party organization, etc.) and who has imposed them (e.g., a Research Ethics Committee or Institutional Review Board, etc.). Please also provide contact information for a data access committee, ethics committee, or other institutional body to which data requests may be sent. b) If there are no restrictions, please upload the minimal anonymized data set necessary to replicate your study findings to a stable, public repository and provide us with the relevant URLs, DOIs, or accession numbers. Please see http://www.bmj.com/content/340/bmj.c181.long for guidelines on how to de-identify and prepare clinical data for publication. For a list of recommended repositories, please see https://journals.plos.org/plosone/s/recommended-repositories. You also have the option of uploading the data as Supporting Information files, but we would recommend depositing data directly to a data repository if possible. Please update your Data Availability statement in the submission form accordingly. 7. In the online submission form, you indicated that [Patient-related data are not available to the public due to privacy issues, but may be obtained from the corresponding author upon reasonable request, subject to approval by the Ethics Committee of Li Huili Hospital, Ningbo Medical Center.]. All PLOS journals now require all data underlying the findings described in their manuscript to be freely available to other researchers, either 1. In a public repository, 2. Within the manuscript itself, or 3. Uploaded as supplementary information.This policy applies to all data except where public deposition would breach compliance with the protocol approved by your research ethics board. If your data cannot be made publicly available for ethical or legal reasons (e.g., public availability would compromise patient privacy), please explain your reasons on resubmission and your exemption request will be escalated for approval.

Reviewers' comments:

Reviewer's Responses to Questions

**Comments to the Author**

1. Is the manuscript technically sound, and do the data support the conclusions?

Reviewer #1: Partly

Reviewer #2: Yes

Reviewer #3: Yes

2. Has the statistical analysis been performed appropriately and rigorously?

Reviewer #1: Yes

Reviewer #2: Yes

Reviewer #3: Yes

3. Have the authors made all data underlying the findings in their manuscript fully available?

Reviewer #1: Yes

Reviewer #2: Yes

Reviewer #3: Yes

4. Is the manuscript presented in an intelligible fashion and written in standard English?

Reviewer #1: No

Reviewer #2: Yes

Reviewer #3: Yes

Reviewer #1: This study presents an intriguing investigation aimed at preoperatively predicting LNM metastasis in CRC patients by using a model consisting of a combination of radiomics and clinical risk factors. The study's innovative approach holds some clinical value, shedding light on the prospect of utilizing radiomics for predicting LNM metastasis in CRC patients. However, in my opinion, the worrisome aspects of this manuscript are as follows:

1) The proportion of patients diagnosed with rectal cancer was unclear, and the tumor deposits were not shown in Table 1, which could affect the N stage of rectal cancer.

2) In the patient selection section, you enrolled the patients who had postoperative pathology reports confirmed CRC, but you did not clarify the type of histopathology. How did you cope with this?

3) The arrows used in Figure 1 were less accurate.

4) You stated that “The images obtained using this tool were also evaluated and confirmed by 2 senior imaging physicians, and their reproducibility was similarly satisfactory”, how did you ensure the reproducibility of your data? It would have been important to examine the variability of results concerning the ROI variation.

5) How did you control colorectal movement during image acquisition? Have you applied some form of motion correction in image post-processing?

6) The graphical abstract showed several methods for model building including SVM, Radom Forest, Decision Tree, etc, have you compared the differences among the methods for model building? That is to say, why did you choose SVM for model building?

7) English is not always clear, English editing is strongly suggested.

Reviewer #2: A VERY WELL WRITTEN ARTICLE. THE AUTHORS HAVE THOROUGHLY FOLLOWED THE AUTHOR GUIDELINES AND COMPLIED VERY WELL WITH THEM.

MeSH BASED KEYWORDS ONLY NEED SOME AMENDMENTS, LYMPHNODE METASTASIS NEED STO BE CORRECTED AND RADIOMICS CHARACTERIZATION OF TUMOURS NEED TO BE REDONE

Reviewer #3: Summary of the research

The stated aim of the study is to develop a model to assess the likelihood of metastatic lymph nodes in colorectal cancer.

The introduction outlined the main problems in preoperative assessment of possible lymph node metastasis and planning surgical treatment for colorectal cancer.

The problems of subjective evaluation of CT by radiologist and evaluation only on the basis of radiological data without taking into account clinical data are identified.

The materials and methods used correspond to the stated objectives in the introduction.

Machine learning is an advanced method in building predictive models based on radiological and clinical data and is suitable for solving the problem posed by the authors.

CT scans were used as radiological data. ROI segmentation (tumor and colorectal region) was performed in manual and semi-automatic mode, which is acceptable, given the complexity of the anatomical region. However, this may be a limiting factor at the stage of reproducing this study, as indicated by the authors in the discussion section.

The ratio between the training and verification sample sizes is optimal (approximately 3:1). Perhaps a larger sample size would have given more weight to the study. This is indicated by the authors in the discussion as a relative limitation of the study.

All materials and methods were described in detail in the relevant section. The names of all programs used are indicated, so reproducing this study is possible.

In general, the data were collected and interpreted correctly. Statistical evaluation of the obtained data was carried out.

Ethical standards were observed in the study.

The obtained results demonstrate the model effectiveness for predicting regional lymph nodes metastases based on complex data (radiomics of the tumor and colorectal zones, clinical data). This corresponds to an AUC of about 0.8, demonstrated both on the training and verification samples.

Thus, the authors propose a solution to an important, clinically significant problem using modern data processing methods. The obtained data allow an optimistic view on the solution to this problem.

Major and minor issues

No major issues were found in this study.

Minor issues:

1. Perhaps there should be more explanation in the text of what is meant by the colorectal region, what are its boundaries. How were the structures segmented in colorectal ROI (lymph nodes, intestine, paracolitic tissue)?

2. Colorectal cancer is a broad term and includes tumors of various parts of the colon and rectum with its anatomical features. Did this fact somehow influence the segmentation of the area of interest? If so, how? If not, why?

3. There is missed reference - line 73.

4. Words are merged in some places in the text.

**Do you want your identity to be public for this peer review?** For information about this choice, including consent withdrawal, please see our Privacy Policy

Reviewer #1: No

Reviewer #2: No

Reviewer #3: No

---

## [Author Response · Author response to Decision Letter 1]

19 Dec 2024

Reviewer #1

1)The proportion of patients diagnosed with rectal cancer was unclear, and the tumor deposits were not shown in Table 1, which could affect the N stage of rectal cancer.

Thank you for your feedback regarding the proportion of patients with rectal cancer and the consideration of tumor deposits.

Proportion of Patients with Rectal Cancer: We have specified that approximately 10% of patients in our study were diagnosed with rectal cancer in the revised manuscript.

Impact of Tumor Deposits: In this study, we primarily utilized lymph node metastasis status from pathology reports to predict preoperative lymph node metastasis. Thus, tumor deposits had minimal impact on our analysis and findings. Additionally, since the majority of cases showed no significant tumor deposits in pathology reports, this variable was not emphasized in Table 1 (Rows 186-190).

We hope this response addresses your queries, and we thank you for your valuable comments.

2)In the patient selection section, you enrolled the patients who had postoperative pathology reports confirmed CRC, but you did not clarify the type of histopathology. How did you cope with this?

We appreciate the reviewers' thorough review of our manuscript. In this study, adenocarcinoma and mucinous adenocarcinoma were identified as the predominant histologic subtypes of colorectal cancer. For patient selection, we specifically included cases of these two subtypes, thus limiting our discussion of other histologic types in the manuscript. This focus ensures relevance and validity in our results, particularly regarding the preoperative prediction of lymph node metastasis (LNM). We appreciate this suggestion and will incorporate relevant annotations in the revised manuscript to enhance clarity and transparency.

3) The arrows used in Figure 1 were less accurate.

We appreciate the feedback regarding the accuracy of the arrows in Figure 1. Figure 1 has been revised to enhance the accuracy and placement of the arrows, ensuring they clearly indicate the intended processes and relationships. Thank you once again for your valuable insights.

4) You stated that “The images obtained using this tool were also evaluated and confirmed by 2 senior imaging physicians, and their reproducibility was similarly satisfactory”, how did you ensure the reproducibility of your data? It would have been important to examine the variability of results concerning the ROI variation.

Response to Reviewer: We thank the reviewers for their valuable comments regarding reproducibility and the importance of ROI variability. Recognizing this, we implemented several steps to validate the reliability of our data. Two senior imaging physicians annotated the colorectal regions in a subset of cases following a standardized protocol. To assess reproducibility, we calculated the intra-group correlation coefficient (ICC) between physician-annotated images and those automatically labeled by our tool, yielding an ICC of 0.994, which demonstrates high concordance. Furthermore, we conducted multiple automated segmentations on the same patient images, achieving an ICC of 0.989, indicating strong reproducibility in image segmentation by the tool.

While these results confirm our study's reproducibility, we acknowledge that further analysis of ROI variability could enhance the robustness of our findings. Future studies will incorporate a more detailed examination of ROI variability to optimize reproducibility further. We appreciate this insightful suggestion and will include these details in the revised manuscript. (Lines 161-167)

5) How did you control colorectal movement during image acquisition? Have you applied some form of motion correction in image post-processing?

Thank you for your insightful question regarding control of bowel movement and motion correction.

Control of bowel motion: During image acquisition, a standardised scanning protocol was followed to minimise bowel movement. Patients were instructed to remain still and in some cases were asked to hold their breath after a deep inhalation to further reduce motion artefacts associated with respiratory movement.

Motion correction in post-processing: We did not apply any specific motion correction algorithms during image post-processing. The imaging sequences and acquisition parameters were optimised to limit motion artefacts and ensure adequate image quality for radiomic analysis. Given the standardised scanning protocol and patient guidance, motion-related issues were minimal in our dataset and no further motion correction was required.

We hope this answers your question and appreciate the opportunity to clarify our methodology.

6) The graphical abstract showed several methods for model building including SVM, Radom Forest, Decision Tree, etc, have you compared the differences among the methods for model building? That is to say, why did you choose SVM for model building?

Response to Reviewer: Thank you for your question regarding the selection of modeling methods. In this study, we initially assessed five machine learning algorithms, including support vector machines (SVM), random forests, and decision trees, to determine the most suitable approach for predicting lymph node metastasis (LNM) in colorectal cancer. We evaluated these methods based on key performance metrics—such as area under the curve (AUC), accuracy, sensitivity, and specificity—to comprehensively assess their predictive capabilities. The Discussion section (lines 312–313) details the performance comparison of these five models, supplemented by Figure 1, which graphically illustrates the AUC differences, providing a visual comparison of model performance on the dataset.

Rationale for Choosing SVM: Among the algorithms tested, SVM consistently demonstrated superior performance across key metrics, particularly AUC. SVM's ability to effectively handle high-dimensional data and complex imaging histologic features was a critical factor in our selection. Additionally, SVM’s low tendency for overfitting and robust classification boundary management further support its suitability for this study.

7)English is not always clear, English editing is strongly suggested.

Response to Reviewer: Thank you for your suggestion regarding the clarity of the English. We have thoroughly reviewed the manuscript and arranged for professional editing to enhance readability and language quality. The revised text has been carefully refined for clarity and precision. We hope these revisions will contribute to improved overall clarity of the manuscript. Once again, we appreciate your valuable feedback.

Reviewer #2

MeSH BASED KEYWORDS ONLY NEED SOME AMENDMENTS, LYMPHNODE METASTASIS NEED STO BE CORRECTED AND RADIOMICS CHARACTERIZATION OF TUMOURS NEED TO BE REDONE

Thank you very much for your constructive comments. In response, we have revised the keywords to align with MeSH standards. Specifically, the term "lymph node metastasis" has been corrected, and all radiological features of the tumor have been rewritten accordingly. In the revised version, all corrected parts are highlighted in red font for clarity.

Reviewer #3

1.Perhaps there should be more explanation in the text of what is meant by the colorectal region, what are its boundaries. How were the structures segmented in colorectal ROI (lymph nodes, intestine, paracolitic tissue)?

Thank you very much for your valuable feedback. We have revised the manuscript to provide a clearer definition of the colorectal region and to detail the segmentation process for the structures within this region of interest (ROI). Below are the specific changes made:

We have added an introduction to the colon region in the Image Acquisition and 3D Segmentation section of the text: The Colorectal Region in this study refers to the anatomical region from the cecum to the rectum and includes the colon, rectum and surrounding paracolic tissue. The boundaries of this region are defined by key anatomical landmarks, including the segments of the colon and the upper and lower ends of the rectum.

In this study, the segmentation of the colorectal region of interest (ROI) focused specifically on delineating the tumor area, excluding structures such as lymph nodes, the intestinal wall, and paracolic tissue. We employed a manual segmentation approach using ITK-SNAP software, where the primary goal was to accurately outline the tumor within the colorectal region while avoiding the inclusion of other structures.

As colorectal tumors are generally well-defined in venous-phase CT images, this manual segmentation method proved to be reliable and reproducible. To ensure the accuracy and consistency of the segmentation, all outlined ROIs were reviewed and confirmed by two senior gastrointestinal radiologists. This process ensured that only the tumor region was included and that other structures, such as lymph nodes or normal intestinal tissues, were not part of the segmented tumor ROI.And we also go into more detail in the manuscript.

We hope this explanation clarifies the segmentation approach used in our study. If you have any further suggestions or questions, we would be happy to address them.

2. Colorectal cancer is a broad term and includes tumors of various parts of the colon and rectum with its anatomical features. Did this fact somehow influence the segmentation of the area of interest? If so, how? If not, why?

Thank you for raising this important point. We acknowledge that colorectal cancer encompasses tumors located in different anatomical regions of the colon and rectum, each with unique characteristics. However, in our study, these anatomical variations did not significantly affect the segmentation of the region of interest (ROI).

To address this, our segmentation method was standardized to focus on the tumor boundaries, regardless of its location within the colorectal region. Since we utilized contrast-enhanced venous phase CT images, tumors in different anatomical locations are generally well-defined, which allowed us to apply a consistent manual segmentation approach across all cases. By concentrating on the well-defined tumor boundaries, we ensured reliable segmentation of both colon and rectal tumors.

3. There is missed reference - line 73.

Thank you for pointing out the omitted reference on line 73. We have now added the appropriate reference to the revised manuscript. The citation has been added to the relevant section to ensure completeness and proper attribution. Thank you for your attention to this detail.

4. Words are merged in some places in the text.

Thank you for your careful review and for pointing out that some words have been merged in the text. We have carefully reviewed the manuscript and corrected all instances where words were unintentionally merged. We have also carefully checked the formatting to ensure clear spacing and readability throughout the document.

We appreciate your attention to detail in helping us improve the clarity of our manuscript.

---

## [Decision Letter · Decision Letter 1]

20 May 2025

Dear Dr. Li,

Thank you for submitting your manuscript to PLOS ONE. After careful consideration, we feel that it has merit but does not fully meet PLOS ONE’s publication criteria as it currently stands. Therefore, we invite you to submit a revised version of the manuscript that addresses the points raised during the review process.

We look forward to receiving your revised manuscript.

Kind regards,

Paolo Aurello

Academic Editor

PLOS ONE

Reviewers' comments:

Reviewer's Responses to Questions

**Comments to the Author**

Reviewer #4: (No Response)

Reviewer #5: All comments have been addressed

2. Is the manuscript technically sound, and do the data support the conclusions?

Reviewer #4: Partly

Reviewer #5: Yes

3. Has the statistical analysis been performed appropriately and rigorously?

Reviewer #4: Yes

Reviewer #5: Yes

4. Have the authors made all data underlying the findings in their manuscript fully available?

Reviewer #4: No

Reviewer #5: Yes

5. Is the manuscript presented in an intelligible fashion and written in standard English?

Reviewer #4: Yes

Reviewer #5: Yes

Reviewer #4: As a new reviewer joining the review process, I apologize if some of my comments below have already been addressed.

The manuscript aims to accurately identify lymphatic metastases using radiomics features extracted from CT images and clinical risk factors. The authors report that the combined clinical and radiomics based model provided an accurate classification of lymphatic metastases.

Please comment if any pre-processing of the CT images was done to ensure that the validation data was harmonized with the training data.

Please comment on the combined model's sensitivity to potential segmentation errors of the tumor and colorectal regions with the automated tool to different noise levels in the images.

Please mention how the regularization parameter for LASSO was chosen for the feature selection of the radiomics features.

It would be interesting to compare the proposed combined model against a basic metric such as the mean, standard deviation, and size of the tumor ROI.

Have the authors considered using a deep-learning neural network for classifying the radiomics features?

Reviewer #5: This multicenter retrospective study aimed to develop a predictive model for lymph node metastasis (LNM) in colorectal cancer (CRC) patients using a combination of 3D radiomics features and clinical risk factors. A total of 349 patients were included, with CT-based radiomic features extracted from both the tumor and colorectal regions, and models were built using support vector machines (SVM). The combined model integrating both radiomics and clinical data (ModelC_3D(R+C)) demonstrated superior performance (AUC: 0.858 training, 0.833 validation) compared to models using radiomics or clinical data alone. The model also incorporated texture features from the colorectal region, which contributed additional predictive value beyond the tumor features. Overall, the study provides evidence that a radiomics–clinical fusion approach significantly improves the accuracy of preoperative LNM prediction and could enhance personalized treatment planning in CRC.

The authors have revised the manuscript according to the reviewers’ suggestions, which has significantly improved the overall quality of the study. However, there are still a few issues that may require further clarification or revision.

1. What was the false positive rate of the combined model in the validation cohort?

2. Could the inclusion of MRI or PET features further improve prediction?

3. Was the model performance statistically compared between training and validation sets?

4. The limitations are acknowledged but could include a note about potential image quality variability across centers.

**Do you want your identity to be public for this peer review?** For information about this choice, including consent withdrawal, please see our Privacy Policy

Reviewer #4: No

Reviewer #5: No

---

## [Author Response · Author response to Decision Letter 2]

2 Jul 2025

Reviewer #4:

1) Please comment if any pre-processing of the CT images was done to ensure that the validation data was harmonized with the training data.

Thank you for your valuable comment. In response to your query regarding the pre-processing of the CT images, we have taken several steps to ensure that the validation data is harmonized with the training data.

1.CT Image Selection: We selected the venous-phase images for both the training and validation datasets, ensuring consistency in the imaging phase across allcases.

2.Image Resizing and Dimensions: All CT images were resized to a consistentdimension of 512x512 pixels to ensure uniformity in image resolution.

3.Image Slice Thickness: A slice thickness of 5mm was applied to all images, both in the training and validation datasets, to maintain uniformity in the 3D reconstruction of the tumors.

4.Image Windowing: During the image segmentation process, we set the Level and Window values to 100 and 300, respectively, for all images. This helped standardize the contrast and brightness of the CT images across the datasets.

These pre-processing steps were applied uniformly to both the training and validation datasets to ensure that the data used for model training and validation were harmonized and comparable. (Lines 133-144)

2) Please comment on the combined model's sensitivity to potential segmentation errors of the tumor and colorectal regions with the automated tool to different noise levels in the images.

Thank you for your thoughtful comment. In our study, we used both manual and automated segmentation techniques to delineate the tumor and colorectal regions, ensuring the accuracy of the tumor area without including surrounding structures such as lymph nodes.

1.Tumor Segmentation: The tumor region was manually segmented using ITK-SNAP software, with venous-phase CT images guiding the process. Given the well-defined appearance of colorectal tumors in CT images, this manual segmentation approach was both effective and reproducible. To ensure segmentation accuracy, two senior gastrointestinal radiologists reviewed and confirmed all manually segmented areas, ensuring no inclusion of non-tumor structures such as lymph nodes or paracolic tissue.

2.Colorectal Region Segmentation: The colorectal region, which includes the colon, rectum, and surrounding paracolic tissue, was segmented using the automated Total Segmentator tool. This tool was chosen for its ability to capture the full extent of the colorectal region, and the resulting segmentations were reviewed and confirmed by two senior imaging physicians. The high intra-group correlation coefficient (ICC) of 0.994 between physician-annotated images and tool-generated segmentations demonstrates strong agreement and high reproducibility.

3.Sensitivity to Segmentation Errors and Noise: While both manual and automated segmentation techniques were rigorously validated for accuracy and reproducibility, we recognize that the presence of noise or segmentation errors could potentially affect model performance. Given that segmentation of the colorectal region is a challenging task, particularly with respect to surrounding tissue structures, we acknowledge that noise and segmentation errors may impact the radiomic features extracted from the images. However, the high ICC values and reproducibility achieved through both manual and automated segmentation methodssuggest that the model is relatively robust to minor variations in segmentation accuracy.

In future work, it would be beneficial to evaluate the sensitivity of the combined model to segmentation errors and image noise through perturbation tests or adversarial noise simulations, which would further assess its robustness under different conditions.

3) Please mention how the regularization parameter for LASSO was chosen for the feature selection of the radiomics features.

Thank you for your valuable comment. In our study, the regularization parameter for LASSO (Least Absolute Shrinkage and Selection Operator) was chosen through a systematic process to optimize the feature selection from the radiomics features.

1.LASSO Regularization Parameter Selection: The optimal value for the regularization parameter (lambda, λ) was determined using cross-validation. Specifically, a 10-fold cross-validation procedure was applied to the training dataset. We tested a range of lambda values, and the one that minimized the mean square error (MSE) across the folds was selected as the optimal parameter. This approach ensures that the model avoids both overfitting (by choosing a lambda thatis too small) and underfitting (by selecting a lambda that is too large).

2.Model Tuning: By performing this process, we ensured that the LASSO method selected the most relevant features while minimizing the inclusion of irrelevant or redundant features, thereby improving the model's generalizability and predictive performance.

In future work, we plan to explore other feature selection methods and furtherfine-tune the LASSO regularization parameter to improve model accuracy and robustness.

3.

4) It would be interesting to compare the proposed combined model against a basic metric such as the mean, standard deviation, and size of the tumor ROI.

Thank you for your suggestion to compare the proposed combined model with basic metrics such as the mean, standard deviation, and size of the tumor ROI. I have conducted the comparison, and the results are as follows:

A comparison of the combined model with the models based on mean, standard deviation, and tumor ROI size shows that the combined model demonstrates superior performance in terms of AUC on both datasets:

Training set (AUC = 0.858 vs. 0.575 vs. 0.541 vs. 0.579)

Validation set (AUC = 0.833 vs. 0.426 vs. 0.467 vs. 0.444)

These results clearly highlight that the combined model significantly outperforms the models that are built directly from basic parameters.(Lines 258-269)

5) Have the authors considered using a deep-learning neural network for classifying the radiomics features?

Thank you for your insightful comment. We appreciate your suggestion regarding the potential use of deep learning for classifying radiomics features.

1.Current Approach and Rationale: In our study, we focused on using traditional machine learning methods, specifically the combined model with LASSO for feature selection and a subsequent classifier, due to their proven interpretability, efficiency, and established success in radiomics-based research. These methods allow for clearer insights into which features contribute most to model performance, which is crucial in medical imaging applications where interpretabilityis key for clinical decision-making.

2.Deep Learning Consideration: While deep learning approaches, especially convolutional neural networks (CNNs), have shown great promise in medical image analysis, they typically require large labeled datasets to achieve high performance. In our study, the dataset size and feature extraction approach led us tofocus on a more interpretable traditional machine learning method. However, weacknowledge that deep learning could potentially further enhance the model’s predictive power, especially in capturing complex patterns within radiomics data.

Future Work: Future Work: Given the rapid advances in deep learning and its increasing success in medical imaging, we plan to explore deep learning techniques in future work. This could involve the use of neural networks for directly analyzing image data or applying deep learning-based feature selection methods before classification. We believe this would complement our existing model and may provide additional predictive power in more complex datasets. As noted in the last section of the manuscript (lines 368-370), we state that "future studies may employ deep learning techniques for radiomics feature extraction, eliminating the need for manual feature design and enabling the capture of more complex data relationships." We will consider this approach in future studies, where a larger dataset or different modalities might be available to leverage the advantages of deep learning techniques.

Reviewer #5

1) What was the false positive rate of the combined model in the validation cohort?

Thank you for your insightful comment.

The false positive rate (FPR) of the combined model in the validation cohort is 0.0833, which indicates that approximately 8.33% of the negative cases in the validation set were incorrectly classified as positive.

As detailed in Supplementary Table 4, we have provided the following key performance metrics for the validation cohort:

AUC: 0.833

Sensitivity: 0.556

Specificity: 0.917

These parameters collectively evaluate the model's performance in predicting preoperative lymph node metastasis (LNM) in CRC patients. The FPR, alongsideAUC, sensitivity, and specificity, provides a comprehensive assessment of the model's diagnostic capability. The false positive rate (FPR) is an essential metric to evaluate the reliability of the model in clinical practice. In our study, the combined model achieved a relatively low FPR of 8.33% in the validation cohort, indicating that the model performed well in distinguishing tumor from non-tumor cases. A low false positive rate is crucial in clinical decision-making, as it helps to reduce unnecessary interventions and avoids the over-diagnosis of tumors, which could lead to unnecessary treatments or tests. Although there were some misclassifications, the model demonstrated a good balance between sensitivity, specificity, and FPR, contributing to its robust performance in predicting preoperative lymph node metastasis (LNM) in CRC patients.(Lines 349-357)

2) Could the inclusion of MRI or PET features further improve prediction?

Thank you for your excellent suggestion. The inclusion of additional imaging modalities such as MRI or PET features could indeed potentially improve the model's prediction accuracy.

1.MRI Features: MRI has been shown to provide superior soft tissue contrast, particularly in detecting lymph node metastases and characterizing tumors with high precision. MRI features, such as texture, shape, and diffusion-weighted imaging (DWI) parameters, could offer complementary information to the radiomic features extracted from CT scans. Integrating MRI features could potentially enhance the model's ability to identify regional lymphatic metastases (LNM),especially in challenging cases where CT images alone might be less conclusive.

2. PET Features: PET imaging is widely recognized for its ability to assess metabolic activity, which is crucial for detecting active tumor regions and lymph node metastases. The inclusion of PET-derived features, such as maximum standardized uptake value (SUVmax), could improve the prediction by providing functional information on the tumor's metabolic activity, which may be a key differentiator between benign and malignant lesions. PET features have been shown to contribute significantly to the accurate staging of cancers, and their integration with radiomics and clinical factors could further refine the model's predictive power.

However, it is important to note that PET is still relatively underutilized in thepreoperative assessment of colorectal cancer patients. PET scanners are not widely available, and only certain hospitals—typically larger tertiary hospitals—are equipped with PET facilities. This makes the inclusion of PET features in routine clinical practice challenging, especially in resource-limited settings. As such,the integration of PET features may need to be limited to specialized centers with the necessary infrastructure.

3. Potential Improvements in Prediction: Combining CT, MRI, and PET features could provide a more comprehensive view of the tumor's structural, functional, and metabolic characteristics, thereby improving the model's performance in predicting LNM. However, the integration of multiple imaging modalities requires advanced techniques in feature extraction, data fusion, and model training. Future studies could explore these multimodal approaches to enhance the accuracy and robustness of preoperative prediction models.

3) Was the model performance statistically compared between training and validation sets?

Thank you for your insightful question. Yes, the performance of all the models was statistically compared between the training and validation sets.

1. AUC Curve Comparison: As shown in Figure 2, we performed a direct comparison of the AUC (Area Under the Curve) for the models trained on the training set versus those validated on the validation set. This visual comparison highlights the discriminative ability of each model in both datasets, ensuring thatthe models are performing consistently across different cohorts.

2. Statistical Comparison of Key Metrics: Additionally, we provided a detailed comparison of key performance metrics, including AUC, sensitivity, and specificity, between the training and validation sets in Supplementary Table 4. This table allows for a more quantitative assessment of the model's performance, showing the AUC, sensitivity, and specificity values for each model in both sets.

These comparisons confirm that the model's performance was consistent and reliable when tested on unseen data in the validation cohort, providing further validation for its generalizability in clinical applications.

4) The limitations are acknowledged but could include a note about potential image quality variability across centers.

Thank you for the suggestion. We acknowledge the importance of image quality variability across different centers, and we agree that this could be a limitation of our study.

While we focused on utilizing standardized imaging protocols to minimize variability, we recognize that differences in scanner models, image resolution, and acquisition techniques across centers may still introduce some degree of variability in the quality of the images used for radiomic feature extraction. This could affect the consistency and reproducibility of the model’s performance when applied to images from different institutions.

We will revise the limitations section of the manuscript to include a note on this potential issue. Specifically, although the two-center design helps validate the model’s generalizability to some extent, broader validation across multiple centers is necessary for more robust validation and assessment. Image quality variability across centers is another limitation to consider. Although standardized imaging protocols were used, differences in scanner models, image resolution, and acquisition techniques could still introduce variability in the quality of images, which may affect the consistency of feature extraction and model performance across centers. .(Lines 359-365)

---

## [Decision Letter · Decision Letter 2]

1 Sep 2025

Radiomics profiling combined with clinical risk factors for preoperative Lymphatic Metastasis prediction in Colorectal cancer: a multicenter study

PLOS ONE

Dear Dr. Li,

Thank you for submitting your manuscript to PLOS ONE. After careful consideration, we feel that it has merit but does not fully meet PLOS ONE’s publication criteria as it currently stands. Therefore, we invite you to submit a revised version of the manuscript that addresses the points raised during the review process.

One reviewer had additional comments that should be addressed in the publication. Please do this in a revision.

If applicable, we recommend that you deposit your laboratory protocols in protocols.io to enhance the reproducibility of your results. Protocols.io assigns your protocol its own identifier (DOI) so that it can be cited independently in the future. For instructions see: https://journals.plos.org/plosone/s/submission-guidelines#loc-laboratory-protocols . Additionally, PLOS ONE offers an option for publishing peer-reviewed Lab Protocol articles, which describe protocols hosted on protocols.io. Read more information on sharing protocols at https://plos.org/protocols?utm_medium=editorial-email&utm_source=authorletters&utm_campaign=protocols.

We look forward to receiving your revised manuscript.

Kind regards,

Gayle E. Woloschak, PhD

Section Editor

PLOS ONE

Journal Requirements:

Additional Editor Comments:

Both reviewers agreed that comments had been addressed. One reviewer had additional comments that appear to be important for the work. The authors should attempt to address those comments.

Reviewers' comments:

Reviewer's Responses to Questions

**Comments to the Author**

Reviewer #4: All comments have been addressed

Reviewer #5: All comments have been addressed

2. Is the manuscript technically sound, and do the data support the conclusions?

Reviewer #4: Yes

Reviewer #5: Yes

3. Has the statistical analysis been performed appropriately and rigorously?

Reviewer #4: Yes

Reviewer #5: Yes

4. Have the authors made all data underlying the findings in their manuscript fully available?

Reviewer #4: No

Reviewer #5: Yes

5. Is the manuscript presented in an intelligible fashion and written in standard English?

Reviewer #4: Yes

Reviewer #5: Yes

Reviewer #4: The authors did a good job addressing my previous comments. Consider making imaging data used here publicly available.

Reviewer #5: This multicenter study developed a machine learning model that integrates CT-based radiomics features from both the tumor and surrounding colorectal region with clinical risk factors to predict preoperative lymph node metastasis (LNM) in colorectal cancer. Using 349 patients (292 training, 57 external validation), the combined model achieved superior performance (AUC = 0.858 training, 0.833 validation) compared with clinical-only or radiomics-only models. The approach demonstrated low false positive rates and good balance of sensitivity and specificity, supporting its clinical utility. Overall, this combined radiomics-clinical model offers a more accurate tool for guiding surgical planning and reducing over- or under-treatment in CRC patient.

However, several sections of the manuscript require revision and clarification before it can be considered for publication.

1. The study used data from two centers, but the external validation set was relatively small (n=57). Could the authors discuss how this may affect the generalizability of their findings and whether larger multicenter validation is planned?

2. The model is based solely on venous-phase CT images. Since MRI and PET offer complementary functional and metabolic information, have the authors considered multimodal imaging integration to further enhance predictive accuracy?

3. The combined model showed good discrimination (AUC >0.83), but sensitivity remained moderate (55.6%). How would this impact clinical decision-making, and what thresholds do the authors recommend for balancing false negatives and false positives in practice?

4. Tumor segmentation was performed manually, while colorectal segmentation used an automated tool. Could the authors elaborate on how segmentation variability may influence radiomics feature stability, and whether robustness testing was performed?

**Do you want your identity to be public for this peer review?** For information about this choice, including consent withdrawal, please see our Privacy Policy

Reviewer #4: No

Reviewer #5: No

---

## [Author Response · Author response to Decision Letter 3]

14 Oct 2025

Comment 1: The study used data from two centers, but the external validation set was relatively small (n=57). Could the authors discuss how this may affect the generalizability of their findings and whether larger multicenter validation is planned?

Response: We thank the reviewer for this constructive comment. We agree that the relatively small external validation cohort (n=57) limits the precision of performance estimates and may affect the generalizability of our findings. While the two-center design provides preliminary external evidence, a small external sample can widen confidence intervals for metrics such as AUC and may not adequately reflect inter-institutional heterogeneity in case-mix and imaging quality. As noted in the Discussion (Lines 359–361), we have acknowledged the limitations of the two-center design and agree that broader multicenter validation is needed. In the revised manuscript, we further clarify the impact of the small external cohort and outline our plan for wider validation. At present, we are engaging in research exchanges with multiple centers and hope to collect and utilize multicenter data for external validation in subsequent work.

Changes in the manuscript: We have added the following text to the Discussion (Limitations) to explicitly acknowledge the effect of the small external cohort and our plan for broader validation: “It should be noted that the external validation cohort was small (n=57), which may lead to wider confidence intervals around performance estimates and therefore warrants cautious interpretation. Although the two-center design provides preliminary evidence of generalizability, broader multicenter validation is necessary to better capture heterogeneity across institutions. At present, we are engaging in research exchanges with multiple centers and hope to collect and utilize multicenter data for external validation in subsequent work.”(Lines 378–384).

Comment 2: The model is based solely on venous-phase CT images. Since MRI and PET offer complementary functional and metabolic information, have the authors considered multimodal imaging integration to further enhance predictive accuracy?

Response: We appreciate this insightful suggestion. We focused on venous-phase CT because it is routinely acquired for preoperative staging in CRC and is widely available across centers, supporting clinical feasibility and generalizability. At study inception, we did consider incorporating MRI- and PET-derived features. However, based on clinical practice in CRC, contrast-enhanced CT is the most commonly available modality, and MRI/PET are not uniformly accessible across hospitals, which further constrained multimodal integration in this retrospective two-center cohort. In addition, standardized MRI/PET data were not consistently available, and inter-center protocol variability would have complicated robust integration and increased the risk of overfitting given the modest sample size.Nonetheless, we agree that MRI (e.g., T2-weighted, DWI/ADC) and PET (e.g., SUV-based metrics) can provide complementary functional and metabolic information that may enhance model performance. We are currently engaging in research exchanges with multiple centers with the aim of collecting datasets from CRC patients who have CT, MRI, and PET imaging available, upon which we will explore developing a higher-performing multimodal predictive model.

Changes in the manuscript: Our model was developed using venous-phase CT alone to prioritize availability and clinical practicality. Although we initially considered incorporating MRI/PET features, in routine CRC practice contrast-enhanced CT is most widely accessible, whereas MRI and PET are not uniformly available across hospitals. Nonetheless, MRI (e.g., T2-weighted, DWI/ADC) and PET (e.g., SUV-based metrics) may provide complementary functional and metabolic information that could enhance performance. We are currently engaging in research exchanges with multiple centers with the aim of collecting datasets from CRC patients who have CT, MRI, and PET imaging available, upon which we will explore developing a higher-performing multimodal predictive model.(Lines 295–303).

Comment 3: The combined model showed good discrimination (AUC >0.83), but sensitivity remained moderate (55.6%). How would this impact clinical decision-making, and what thresholds do the authors recommend for balancing false negatives and false positives in practice?

Response: We appreciate the reviewer’s important question. While the model achieved strong overall discrimination (AUC >0.83) and a low false positive rate (FPR) of 8.33% in the validation cohort—indicating high specificity (~91.7%) and reduced risk of over-treatment—the sensitivity at the chosen operating point was moderate (55.6%). This implies a non-negligible risk of false negatives (missed LNM) and highlights that the model should complement, rather than replace, clinical judgment and conventional imaging.

Threshold selection should be tailored to clinical priorities. When the consequence of missing LNM is high (e.g., management would materially change), lowering the decision threshold to increase sensitivity is advisable, accepting a higher FPR. Conversely, when minimizing over-treatment is paramount, a higher threshold can be used to favor specificity. For routine use, a balanced operating point (e.g., near the Youden index) can provide a reasonable trade-off. We recommend that institutions calibrate thresholds to local case-mix and resource considerations, guided by model calibration and decision-curve analysis to maximize net benefit. Probability-based risk stratification (e.g., low-, intermediate-, and high-risk bands) may offer more nuanced support than a single binary cut-off.

Changes in the manuscript: We have revised the paragraph in the Discussion addressing false positives to incorporate the reviewer’s concerns about sensitivity and thresholding. The updated paragraph reads: “The false positive rate (FPR) is an essential metric to evaluate the reliability of the model in clinical practice. In our study, the combined model achieved a relatively low FPR of 8.33% in the validation cohort, indicating that the model performed well in distinguishing tumor from non-tumor cases and yielding high specificity (~91.7%) alongside good overall discrimination (AUC >0.83). However, the sensitivity at the chosen operating point remained moderate (55.6%), implying a risk of false negatives (missed LNM) that warrants cautious interpretation, particularly when LNM status could alter management. This operating profile is conservative—helping to reduce unnecessary interventions and avoid over-diagnosis—yet it underscores the need to tailor decision thresholds to clinical priorities: lowering the threshold to increase sensitivity when avoiding under-treatment is critical, and raising the threshold to emphasize specificity when minimizing over-treatment is paramount. For routine use, a balanced operating point (e.g., near the Youden index) may provide an appropriate trade-off. We recommend site-specific calibration and the use of calibration assessment and decision-curve analysis to identify thresholds that maximize net benefit. Probability-based risk stratification (e.g., defining low-, intermediate-, and high-risk bands) may further support nuanced decision-making. Although some misclassifications were observed, the model demonstrated a favorable balance among sensitivity, specificity, and FPR, and is intended to complement—rather than replace—clinical judgment in predicting preoperative LNM in CRC patients.”(Lines 358–376).

Comment 4: Tumor segmentation was performed manually, while colorectal segmentation used an automated tool. Could the authors elaborate on how segmentation variability may influence radiomics feature stability, and whether robustness testing was performed?

Response: We appreciate the reviewer’s important question. Segmentation variability can affect the stability of radiomics features, particularly boundary-sensitive shape and texture descriptors. For the tumor ROI, we used venous-phase CT to enhance contrast between the tumor and surrounding tissues and performed manual delineation in ITK-SNAP with a rigorous quality-control workflow: two senior gastrointestinal radiologists reviewed and confirmed all tumor contours to ensure complete lesion coverage and exclusion of non-tumor structures (normal bowel, lymph nodes, paracolic tissue). These procedures are described in the manuscript (Lines 153–156) and were intended to minimize intra- and inter-observer variability. For the colorectal region, we employed an automated tool based on Total Segmentator and validated its agreement with physician annotations. The automated contours showed excellent concordance with expert-defined regions (ICC = 0.994), and repeated automated segmentations of the same patient’s image demonstrated high reproducibility (ICC = 0.989), as detailed in the manuscript (Lines 164–168). These findings suggest that features derived from the colorectal region are likely to be stable with respect to segmentation variability.

Changes in the manuscript: We have expanded the Discussion to further explain how segmentation variability may influence radiomics feature stability and to place our workflow and validation results in context. The added text reads: “Segmentation variability can propagate to radiomics features via boundary placement, ROI size/shape, and the inclusion or exclusion of adjacent tissues. Boundary shifts predominantly affect shape metrics (e.g., volume, surface area, sphericity/compactness), boundary-sensitive texture features (e.g., GLCM contrast/entropy, GLRLM/GLSZM nonuniformity), and high-frequency wavelet or Laplacian-of-Gaussian features, whereas global intensity histogram features may be less sensitive to small boundary changes. Partial-volume effects at the tumor–bowel interface and inadvertent inclusion of peritumoral tissues can alter heterogeneity descriptors and impact downstream model outputs. To mitigate variability, tumor ROIs were manually delineated on venous-phase CT and reviewed by two radiologists; automated colorectal segmentation based on Total Segmentator showed excellent agreement with expert annotations (ICC = 0.994) and good repeatability (ICC = 0.989), supporting the stability of features in this region. In future work, we will explore deep learning–driven tumor auto-segmentation to further reduce operator dependence and enhance cross-center stability.”(Lines 390–403).

---

## [Decision Letter · Decision Letter 3]

27 Oct 2025

Dear Dr. Li,

**One reviewer had no changes and accepted the work. The other had some changes; most significant among these are the statistical issues. Please address all of these in a resubmission.**

We look forward to receiving your revised manuscript.

Kind regards,

Gayle E. Woloschak, PhD

Section Editor

PLOS ONE

**Journal Requirements:**

**Additional Editor Comments:**

One reviewer accepted the work, the other suggested major revisions. When I have reviewed these, I think they can all be addressed with modifications to the manuscript. Please address these in a revision.

Reviewers' comments:

Reviewer's Responses to Questions

**Comments to the Author**

Reviewer #4: (No Response)

Reviewer #5: All comments have been addressed

2. Is the manuscript technically sound, and do the data support the conclusions?

Reviewer #4: (No Response)

Reviewer #5: Yes

3. Has the statistical analysis been performed appropriately and rigorously?

Reviewer #4: (No Response)

Reviewer #5: N/A

4. Have the authors made all data underlying the findings in their manuscript fully available?

Reviewer #4: (No Response)

Reviewer #5: Yes

5. Is the manuscript presented in an intelligible fashion and written in standard English?

Reviewer #4: (No Response)

Reviewer #5: Yes

**Reviewer #4**:  (No Response)

**Reviewer #5:**  This multicenter retrospective study developed a machine learning (SVM-based radiomics model integrating 3D venous-phase CT features of both tumor and colorectal regions with clinical risk factors to predict preoperative lymph node metastasis (LNM) in colorectal cancer (CRC). A total of 349 patients (292 training, 57 external validation) were analyzed. The combined model (clinical + 3D[R+C]) achieved the best predictive performance (AUC = 0.858 in training; 0.833 in validation), outperforming clinical-only or radiomics-only models. The study concludes that combining radiomics and clinical features enhances diagnostic accuracy for LNM, providing a noninvasive tool to improve preoperative CRC treatment planning and reduce over- or under-treatment risks. The revised manuscript has been substantially improved; however, some parts still require revision before it can be accepted for publication.

1. Were all CT scans obtained using the same scanner parameters, slice thickness, and reconstruction algorithms across centers? Variability in imaging protocols can significantly affect radiomic feature reproducibility, so this should be discussed or controlled for statistically.

2. The manuscript does not specify whether intra- and inter-observer reproducibility was tested for ROI segmentation. Radiomics models are sensitive to manual delineation; therefore, inclusion of intra-class correlation coefficient (ICC) analyses would increase confidence in feature robustness.

3. Which radiomic features contributed most strongly to LNM prediction? Interpretation of these features in relation to tumor biology—such as heterogeneity or vascularity—would improve readability for clinicians unfamiliar with radiomics.

4. Include 95% confidence intervals for AUC values.

what does this mean?). If published, this will include your full peer review and any attached files.

**Do you want your identity to be public for this peer review?** For information about this choice, including consent withdrawal, please see our Privacy Policy

Reviewer #4: No

Reviewer #5: No

---

## [Author Response · Author response to Decision Letter 4]

11 Dec 2025

1. Were all CT scans obtained using the same scanner parameters, slice thickness, and reconstruction algorithms across centers? Variability in imaging protocols can significantly affect radiomic feature reproducibility, so this should be discussed or controlled for statistically.

Response:Thank you for your feedback on our manuscript. In our study, we first performed 0-1 normalization on the extracted radiomic features to reduce variability arising from different scanning conditions. Specifically, we employed the following approach:

Purpose of 0-1 Normalization: Different scanning devices and parameters can lead to variations in the range of feature values, which can affect the assessment of tumor characteristics. 0-1 normalization effectively eliminates these differences by standardizing the scale of feature values, allowing for valid comparisons within the same range.

Method of Implementing 0-1 Normalization: We used the following formula to transform each feature value into a scale between 0 and 1:

X=X−min(X)max(X)−min(X)

This approach ensures that all feature values are rescaled to the range of [0, 1].

Analysis of Normalized Features: The normalized features will be used for subsequent statistical analyses. This will enable us tomore accurately evaluate the stability of features under different scanning conditions and their impact on model performance.

By implementing 0-1 normalization, we have established a solid foundation for subsequentcomparisons and analyses, ensuring the comparability and consistency of radiomic features extracted under different scanning conditions.

Changes in the manuscript: We have added the following text to the statistical analysis section to clarify how we addressed the variations in images caused by different scanning devices across centers:

"First, we divided the patient data into a training set and an external validation set. After extracting features using radiomic methods, we performed 0-1 normalization on all extracted features due to significant differences in the ranges of radiomic feature values resulting from various scanning devices and parameter settings. These differences could potentially impact the evaluation of tumor characteristics. This normalization process is a crucial step to ensure the comparability of features and the reliability of the analysis results."(Lines 204-209).

2. The manuscript does not specify whether intra- and inter-observer reproducibility was tested for ROI segmentation. Radiomics models are sensitive to manual delineation; therefore,inclusion of intra-class correlation coefficient (ICC) analyses would increase confidence in feature robustness.

Response: We thank the reviewer for highlighting the importance of assessing intra- and inter-observer reproducibility for ROI segmentation. In response to this valuable feedback, we conducted intra-class correlation coefficient (ICC) analyses. Specifically, we calculated the ICC for ROI segmentations performed by the same observer on the same patient one day apart, which yielded an ICC of 0.973, indicating excellent intra-observer reliability. Additionally, we assessed the ICC between two different observers segmenting the ROI for the same patient, which resulted in an ICC of 0.972, demonstrating strong inter-observer agreement.We believe that these results affirm the robustness of our ROI segmentation process and enhance the reliability of the radiomic features extracted in our study.

Changes in the manuscript: In the "Image Acquisition and 3D Segmentation" section, we have added the following text to explicitly report the ICC analyses:

“ To ensure consistency and accuracy, two senior gastrointestinal radiologists reviewed and confirmed all manually segmented tumor regions, verifying segmentation accuracy and ensuring that no additional structures, such as lymph nodes or paracolic tissue, were included in the tumor ROI. To evaluate the reproducibility of ROI segmentation, we conducted intra-class correlation coefficient (ICC) analyses. The ICC for segmentation performed by the same observer on the same patient one day apart was 0.973, indicating excellent intra-observer reliability. Furthermore, the ICC calculated between two different observers segmenting the same patient’s ROI was 0.972, demonstrating strong inter-observer agreement. These results further support the reliability of our radiomic feature extraction process. ” (Lines 155-164).

3. Which radiomic features contributed most strongly to LNM prediction? Interpretation of these features in relation to tumor biology—such as heterogeneity or vascularity—would improve readability for clinicians unfamiliar with radiomics.

Response: We thank the reviewer for this insightful comment. We agree that understanding which radiomic features contribute most significantly to lymph node metastasis (LNM) prediction is crucial for clinical interpretation. we have included a SHAP (SHapley Additive exPlanations) plot in our manuscript.  In our analysis, we identified two key features that played a substantial role: original_shape_Compactness from the 3D(R) imaging features and log-sigma_3_mm_3D_gldm_SmallDependenceHighGrayLevelEmphasis from the 3D(C) imaging features.

Interpretation of Key Features:

The original_shape_Compactness is indicative of the tumor's geometric properties. A higher compactness suggests a more homogeneous tumor structure, which may correlate with a lower likelihood of metastasis. In contrast, irregular shapes often reflect increased tumor heterogeneity, potentially indicating a higher risk of lymph node involvement. This relationship underscores the importance of tumor morphology in understanding metastatic behavior.

The log-sigma_3_mm_3D_gldm_SmallDependenceHighGrayLevelEmphasis feature highlights the distribution of high gray-level values within the tumor. This characteristic can signify areas of increased vascularity, which are essential for tumor growth and metastasis. Enhanced vascularity may facilitate the spread of tumor cells, thereby increasing the risk of lymph node metastasis. By elucidating these associations, we aim to provide clinicians with a clearer understanding of how specific imaging features relate to tumor biology.

Changes in the manuscript: We have added the following text to the Discussion to elaborate on the contributions of these features to LNM prediction:

“Based on the SHAP values obtained from our analysis, we found that the original_shape_Compactness and log-sigma_3_mm_3D_gldm_SmallDependenceHighGrayLevelEmphasis features significantly contribute to our predictive model for lymph node metastasis. The former reflects tumor geometric properties, suggesting that higher compactness may correlate with lower metastatic potential. In contrast, the latter emphasizes high gray-level areas that are indicative of increased vascularity, thus highlighting the importance of tumor biology in predicting lymph node involvement. By elucidating these associations, we aim to provide clinicians with a clearer understanding of how specific imaging features relate to tumor biology.” (Lines 333–341).

4. Include 95% confidence intervals for AUC values.

Thank you for your suggestion to include 95% confidence intervals (CIs) for the AUC values. We have incorporated the corresponding 95% confidence intervals for all AUC values in the manuscript, and they are highlighted in red for clarity.

---

## [Decision Letter · Decision Letter 4]

21 Dec 2025

Radiomics profiling combined with clinical risk factors for preoperative Lymphatic Metastasis prediction in Colorectal cancer: a multicenter study

PONE-D-24-31123R4

Dear Dr. LI:

We’re pleased to inform you that your manuscript has been judged scientifically suitable for publication and will be formally accepted for publication once it meets all outstanding technical requirements.

Kind regards,

Gayle E. Woloschak, PhD

Section Editor

PLOS One

Additional Editor Comments (optional):

Reviewers' comments:

Reviewer's Responses to Questions

**Comments to the Author**

Reviewer #5: All comments have been addressed

2. Is the manuscript technically sound, and do the data support the conclusions?

Reviewer #5: Yes

3. Has the statistical analysis been performed appropriately and rigorously?

Reviewer #5: Yes

4. Have the authors made all data underlying the findings in their manuscript fully available?

Reviewer #5: Yes

5. Is the manuscript presented in an intelligible fashion and written in standard English?

Reviewer #5: Yes

Reviewer #5: The authors have satisfactorily revised the manuscript in accordance with the reviewers’ recommendations. As no further comments are needed. therefore, the reviewers have accepted the manuscript for publication in PLOS One.

**Do you want your identity to be public for this peer review?** For information about this choice, including consent withdrawal, please see our Privacy Policy

Reviewer #5: No

---

## [Editor Report · Acceptance letter]

PONE-D-24-31123R4

PLOS One

Dear Dr. Li,

I'm pleased to inform you that your manuscript has been deemed suitable for publication in PLOS One. Congratulations! Your manuscript is now being handed over to our production team.

Kind regards,

on behalf of

Dr. Gayle E. Woloschak

Section Editor

PLOS One